# InferSpec: Adaptive Inference-Time Compute with Ensemble Verifier-Guided Speculative Decoding for Efficient Reasoning

## Abstract

Large language models (LLMs) are effective at multistep reasoning, but suffer from high inference costs, making efficient deployment challenging. Although speculative decoding (SD) offers latency reductions by letting a lightweight draft propose tokens that a stronger target verifies, yet its token-centric nature admits subtle flaws in intermediate steps to propagate, ultimately producing incorrect final output. The existing literature, such as reward-guided SD, rely on external pre-trained reward models, which increase latency and limit generalizability. To overcome this limitation, we propose InferSpec, a mathematically grounded, verification-aware framework for adaptive inference-time compute. At each step, InferSpec samples multiple draft candidates and applies a self-consistency selector to choose a representative one. It then evaluates the selected step using two model-internal criteria: (i) Attention-Based Grounding Verification (ABGV), which computes grounding scores from attention rollout matrices to ensure attribution to inputs or prior steps, and (ii) Log-Probability-Based Verification (LPBV), which bounds token-level confidence. These signals form a weighted ensemble score with formal guarantees that only grounded, high-confidence steps are accepted; uncertain steps escalate to the target model, allocating compute selectively. Experiments on MATH500, GSM8K, Gaokao-2023-En, and Olympiad-Bench show that InferSpec improves accuracy by 3.6% while reducing latency by ∼11%, consistently outperforming both standard SD and reward-guided SD.

## 1 Introduction

Large language models (LLMs) have demonstrated a remarkable ability to solve complex multi-step reasoning problems across domains such as mathematics and knowledge-intensive tasks Brown et al. (2020); Team et al. (2024); Hurst et al. (2024). However, their practical deployment is constrained by high inference costs, which limit scalability and real-time applicability Patterson et al. (2021). *Reducing inference overhead without sacrificing accuracy has therefore become a central research challenge* Frantar et al. (2023); Xu et al. (2024); Lin et al. (2024).

Speculative decoding (SD) Leviathan et al. (2023) has emerged as a promising solution to accelerate inference, where a lightweight draft model generates candidate tokens, and a stronger target model verifies them. By offloading much of the token generation process to the smaller draft model, SD achieves significant latency reductions compared to decoding with the target model alone. Despite these gains, SD remains inherently token-centric, leading to critical limitations in reasoning tasks. Its strict unbiasedness requirement often rejects semantically correct draft tokens that have low probability under the target model, resulting in wasted computation and reduced efficiency Bachmann et al. (2025); Holtzman et al. (2020). This rigidity limits speedups and makes it less effective for multi-step tasks such as math and coding.

Recent extensions of SD attempt to address this limitation. For example, reward-guided speculative decoding (RSD) Liao et al. (2025) introduces external pre-trained reward models (PRMs) to verify the correctness of the draft output. Although effective in improving reliability, it incurs substantial drawbacks. First, reliance on external verifiers significantly increases latency and compute overhead.

Second, pre-trained reward models are often specialized to specific domains or tasks, making them difficult to generalize across diverse reasoning tasks.

This naturally leads to the central question driving our work: *How can we design a speculative decoding framework that maintains accuracy in multi-step reasoning tasks while remaining cost-efficient and scalable, without relying on external verifier models?*

In this paper, we present **INFERSPEC**, a mathematically grounded, verification-aware framework for adaptive inference-time compute allocation. The key intuition behind INFERSPEC is:

- *Accuracy preservation:* Mitigate error propagation by ensuring that only trusted intermediate outputs are accepted, thereby safeguarding correctness throughout the reasoning chain.
- *Efficiency:* Enable lightweight, cost-effective verification without relying on large external verifiers, thus reducing latency.

INFERSPEC integrates two lightweight verifiers derived directly from the model itself: (i) *Attention-based grounding verification*, which checks whether the generated step is properly grounded in the input context or previously validated steps, and (ii) *Log-probability-based verification*, which ensures confidence at the token level. These complementary signals are combined into an *ensemble verifier* that adaptively decides whether to accept draft outputs or invoke the target model. Furthermore, we introduce a novel *self-consistency selector* that identifies the most semantically consistent reasoning step from multiple sampled draft candidates. To summarize, our key contributions are:

1. We propose INFERSPEC, a novel framework that integrates model-internal verifiers with adaptive inference-time compute allocation, improving reliability without the need for external reward models.
2. We introduce a novel *self-consistency selector* that identifies the most representative reasoning step from multiple sampled draft candidates.
3. Extensive experiments on various reasoning benchmarks show that INFERSPEC improves accuracy by up to 3.6% while reducing latency by ∼11% compared to state-of-the-art methods, establishing it as both effective and efficient for real-world LLM deployment.

## 2 RELATED WORK

**Speculative Decoding.** Speculative decoding accelerates inference by letting a lightweight draft model propose tokens that a larger target model verifies in parallel Leviathan et al. (2023); Li et al. (2024); Chen et al. (2024c; 2023); Zhang et al. (2024); Stern et al. (2018); Xia et al. (2024); Sun et al. (2024). Variants include tree-based speculation Chen et al. (2024b); Sun et al. (2023); Fu et al. (2024); Miao et al. (2024) to increase acceptance, self-speculative decoding that leverages parts of the base model Zhang et al. (2024); Elhoushi et al. (2024), and CTC-based drafting Wen et al. (2024) to improve sequence quality. Methods like LayerSkip Elhoushi et al. (2024) and Draft-on-the-Fly Metel et al. (2024) further explore adaptive or early-exit strategies. SpecReason Pan et al. (2025) performs speculative reasoning with the target model as a critic that scores semantic utility via a single-token threshold rule. INFERSPEC, in contrast, combines multi-sample self-consistency with an ensemble verifier, enabling stronger filtering of plausible-but-ungrounded steps. RSD Liao et al. (2025) incorporates process reward models (PRMs) to guide speculative reasoning at the step level. INFERSPEC differs by keeping the standard draft-target pipeline but replacing external verifiers with lightweight, model-internal signals for step-level evaluation.

**Reward Models on Reasoning.** Reward models are used to provide feedback for choosing the correct reasoning path Zhou et al. (2025); Wang et al. (2024); Chen et al. (2024a). Outcome reward models (ORMs) Dong et al. (2024); Yu et al. (2024) score final answers, while process reward models (PRMs) Lightman et al. (2023) assess intermediate steps. The advancement of reward models has brought increasing attention to scaling test-time compute Snell et al. (2024). They enable strategies like Best-of-N sampling Dong et al. (2023); Cobbe et al. (2021); Brown et al. (2024), tree search Yao et al. (2023); Qi et al. (2024); Chen et al. (2024a), and reward-guided inference such as RSD Liao et al. (2025) or SPECS Cemri et al. (2025). These improve reasoning quality but add latency and reliance on external verifiers. In contrast, INFERSPEC leverages an ensemble of internal confidence and grounding signals, avoiding external PRMs while improving multi-step reasoning accuracy.

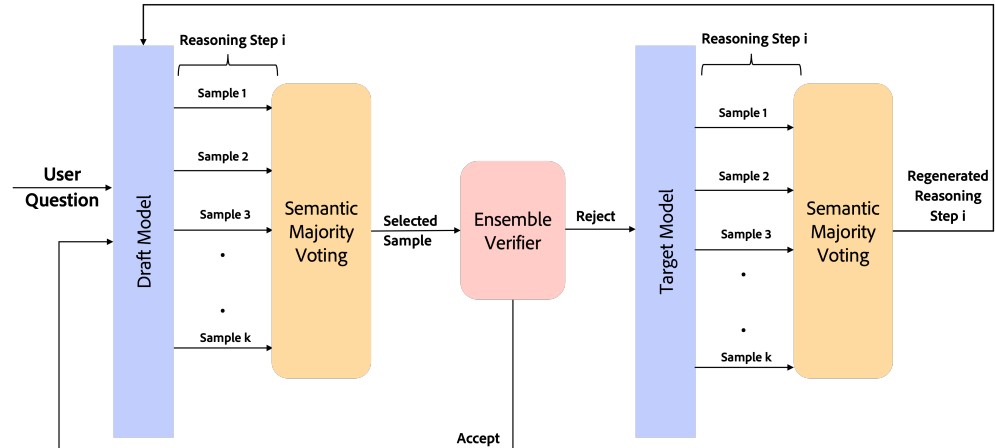

Figure 1: Architectural overview of the INFERSPEC framework

# 3 OUR PROPOSED APPROACH

In this section, we present our proposed novel framework for speculative decoding with inference time compute, in which we introduce an ensemble verifier that integrates attention-based grounding verification (Section 3.1) with probability-based signals (Section 3.2). This formulation enables efficient reasoning without reliance on external verifiers while maintaining interpretability and robustness. We then describe how our approach combines the inference time compute with the ensemble-guided acceptance criteria (Section 3.3)), resulting in a method we call INFERSPEC. Figure 1 outlines the high-level architectural overview of our proposed framework.

## 3.1 ATTENTION-BASED GROUNDING VERIFICATION (ABGV)

We introduce Attention-Based Grounding Verification (ABGV) as a mechanism to assess whether each output token (i.e. full reasoning step in our scenario) generated by a language model is sufficiently grounded in the input context or the previously generated steps. Unlike approaches that rely on external verifiers or auxiliary models, ABGV directly leverages the internal attention matrices of the model itself, enabling efficient and scalable verification. The key intuition is that a correctly grounded output should exhibit strong attention alignment with the most relevant input tokens or validated prior steps, thereby reflecting faithful attribution rather than spurious correlations.

Let an input prompt be denoted as $x$, and the language model generates an output sequence $y_i = (y_{i,1}, y_{i,2}, \ldots, y_{i,T})$ at each step $i$. At each generation step, the model produces multilayer multihead attention matrices for each layer $l$ and head $h$: $A^{(l,h)} \in \mathbb{R}^{(t_{\text{input}}+t_{\text{output}}) \times (t_{\text{input}}+t_{\text{output}})}$.

To compute cumulative attribution from input tokens to an output token, we use the well-known *attention rollout* mechanism, which recursively multiplies attention matrices across layers to show the total influence of each input token on the final output. Formally, let $A^{(l)}$ denote the attention matrix averaged over the heads in layer $l$. The rollout matrix $R$ is computed as: $R = A^{(L)} A^{(L-1)} \cdots A^{(1)}$.

For each output token $y_{i,t}$, the distribution over the input tokens is given by the row $R_{y_{i,t}}$ of the rollout matrix, normalized to sum to 1. Let $\mathcal{I}$ denote the set of input tokens (including prior reasoning steps). The grounding score for token $y_{i,t}$ is defined as:

$$G(y_{i,t}) = \sum_{j \in \mathcal{I}} R_{y_{i,t}}[j]$$

Here, $R_{y_{i,t}}[j]$ denotes the attention weight of the token $y_{i,t}$ to the input token $j$. A higher grounding score indicates a stronger reliance on the input context. We adopt a stricter criterion by taking the minimum token grounding score across the reasoning step $y_i$: $G_{\text{min-step}} = \min_t G(y_{i,t})$,

which ensures that every token in the generated reasoning step $y_i$ is sufficiently grounded, thereby preventing ungrounded tokens from being masked by averaging.

**Memory-Efficient Design:** A naive implementation would require storing attention matrices from all layers, which could become memory-intensive for larger models and longer outputs. To ensure practical scalability, ABGV employs two lightweight design choices:

- **Layer subset:** we store attention matrices from only the last 3 layers, which we find sufficient for grounding quality (Figure 3(a) shows minimal loss in verification performance).
- **Head sparsification:** we discard entries below 0.01 in attention heads, significantly reducing memory footprint with negligible effect on grounding fidelity (see Figure 3(b)).

## 3.2 LOG PROBABILITY-BASED VERIFICATION (LPBV)

We introduce log-probability-based verification (LPBV) as a complementary mechanism to assess the reliability of the full reasoning step generated by a language model. LPBV relies on the model's own predictive confidence, as reflected in the conditional logarithmic probability of the tokens generated. The key intuition is that faithfully generated and reliable output should be assigned a relatively high logarithmic probability under the model's next-token distribution, while ungrounded tokens are often associated with predictions of low probability. For each token $y_{i,t}$, of the reasoning step $y_i$, the model produces a conditional probability given the input $x$ and the prior steps: $p(y_{i,t} \mid x, y_{i,<t})$, from which we compute the logarithmic probability score:

$$L(y_{i,t}) = \log p(y_{i,t} \mid x, y_{i,<t})$$

At each step level, a stricter criterion is applied by taking the minimum log probability across tokens, ensuring that no token is assigned disproportionately low confidence: $L_{\text{min-step}} = \min_t L(y_{i,t})$ .

## 3.3 INFERSPEC: ADAPTIVE INFERENCE-TIME COMPUTE WITH ENSEMBLE VERIFIER-GUIDED SPECULATIVE DECODING

We propose INFERSPEC, an ensemble verifier-guided speculative decoding framework that augments speculative decoding with principled verification at the step level. At each reasoning step, INFERSPEC evaluates draft outputs using two lightweight, model-internal signals: (i) *Log Probability-Based Verification (LPBV)*, which enforces token-level confidence by measuring predictive likelihoods, and (ii) *Attention-Based Grounding Verification (ABGV)*, which ensures that every generated token is properly attributed to the input or previously validated steps via attention rollout. These complementary criteria are combined into a unified ensemble score that carries **formal guarantees**: only steps that are simultaneously confident and grounded are accepted, while uncertain steps trigger recomputation with the target model. In doing so, INFERSPEC mitigates error cascades (which is common in speculative decoding), thus improving reasoning reliability while preserving efficiency. In each reasoning step $i$, INFERSPEC proceeds as follows:

**A. Generate Draft Step:** The draft model $m$ samples $k$ candidate reasoning steps $\{\hat{y}_i^{(1)}, \ldots, \hat{y}_i^{(k)}\}$ conditioned on the input prompt and previously accepted steps. To identify the most consistent candidate from these possibilities $k$, we propose the *self-consistency selector* (see Section 3.4 for more details), which selects the step $\hat{y}_i^{j^*}$ that is maximally consistent with the other candidates $k-1$.

**B. Compute Verification Scores:** For the selected step, the ensemble verifier computes both the logarithmic probability-based score $L(\hat{y}_i^{j^*})$ and the grounding score $G(\hat{y}_i^{j^*})$. Before aggregation, both scores are scaled to a comparable range using Min-Max normalization.

**C. Apply Acceptance Criterion:** The ensemble verifier combines normalized scores through a weighted aggregation to determine acceptance. If the criterion is satisfied, $\hat{y}_i^{j^*}$ is accepted; otherwise, the target model $M$ is invoked to sample $k$ candidate steps $\{y_i^{(1)}, \ldots, y_i^{(k)}\}$ to reduce stochastic variance and improve reliability. Since even the target model may occasionally produce inconsistent reasoning. *Self-consistency selector* is again applied to select the most consistent step $y_i^{j^*}$.

**D. Repeat Until Termination:** This process continues until the model generates an end-of-sequence (EOS) token or the sequence reaches the maximum length $N$.
Algorithm 1 outlines the key steps involved in the proposed approach INFERSPEC.

Analysis of the computational complexity of INFERSPEC is provided in Appendix A.3.

---

**Algorithm 1:** INFERSPEC

---

**Input:** Prompt $x$, draft model $m$, target model $M$, log prob function $L(\cdot)$, grounding score function $G(\cdot)$,
log prob range $[\ell_{\min}, \ell_{\max}]$, grounding range $[g_{\min}, g_{\max}]$, weight $\beta$, acceptance threshold $\tau$, EOS
token $s$, max length $N$, samples per step $k$

**Output:** Response $y_{1:i}$

1   Initialize $y_{1:0} \leftarrow$ ""

2   **for** $i \leftarrow 1$ **to** $N - 1$ **do**

3      Sample $k$ draft candidates $\{\hat{y}_i^{(1)}, \ldots, \hat{y}_i^{(k)}\} \leftarrow m(x, y_{1:i-1})$

4      Select draft step $\hat{y}_i^{j*} \leftarrow$ Self-Consistency Selector($\{\hat{y}_i^{(j)}\}_{j=1}^k$)

5      Compute min log prob $\ell_i \leftarrow L(\hat{y}_i^{j*})$

6      Compute min grounding score $g_i \leftarrow G(\hat{y}_i^{j*})$

7      Normalize: $\tilde{\ell}_i = \dfrac{\ell_i - \ell_{\min}}{\ell_{\max} - \ell_{\min}}, \tilde{g}_i = \dfrac{g_i - g_{\min}}{g_{\max} - g_{\min}}$

8      Compute ensemble verifier score: $r_i \leftarrow \beta \cdot \tilde{\ell}_i + (1 - \beta) \cdot \tilde{g}_i$

9      **if** $r_i \geq \tau$ **then**

10         Accept draft step $y_i \leftarrow \hat{y}_i^{j*}$

11      **else**

12         Sample $k$ target candidates $\{y_i^{(1)}, \ldots, y_i^{(k)}\} \leftarrow M(x, y_{1:i-1})$

13         Select target step $y_i^{j*} \leftarrow$ Self-Consistency Selector($\{y_i^{(j)}\}_{j=1}^k$)

14         $y_i \leftarrow y_i^{j*}$

15      **if** $s \in y_i$ **then**

16         **break**

---

### 3.4   SELF-CONSISTENCY SELECTOR TO IDENTIFY THE MOST CONSISTENT CANDIDATE

To identify the most consistent reasoning step among a set of $k$ sampled candidates (either by draft or target), we propose the *self-consistency selector*, based on Zhu et al. (2025). The underlying intuition is that a consistent candidate should exhibit strong agreement with the other candidates, rather than being an outlier. Formally, each candidate $y^{(j)}$ is encoded in a normalized embedding $e^{(j)}$ using a pre-trained sentence transformer $\mathcal{E}$. The (cosine) similarities are then calculated in pairs between the candidates, yielding a similarity matrix $S \in \mathbb{R}^{k \times k}$, which is further normalized row-wise using softmax to obtain $\tilde{S}$. For each candidate $y^{(j)}$, we calculate its self-alignment score $d_j = \tilde{S}_{jj}$, which measures the degree to which the candidate aligns with itself relative to the others. Candidates that are semantically consistent with the rest of the set distribute their similarity mass across multiple peers, producing a low $d_j$, while outliers or less consistent candidates concentrate the similarity primarily on themselves, resulting in a high $d_j$. Thus, candidates with lower $d_j$ are more representative of the set, and our approach selects the candidate with the minimum self-alignment score: $j^* \leftarrow \arg\min_j d_j$. Algorithm 2 describes our novel self-consistency selector.

### 3.5   FORMAL GUARANTEES

We now present formal guarantees for the proposed INFERSPEC algorithm.

**Lemma 1** (Soundness Guarantee). *Let $\mathcal{C}$ denote the set of correct reasoning steps, $\tilde{\ell}_i$ be the logarithmic probability signal, and $\tilde{g}_i$ be the attention-grounding signal. For for any $\alpha \in [0, 1]$, $\epsilon_\ell \in [0, 1]$ and $\epsilon_g \in [0, 1]$, assume that $\Pr\left[\tilde{\ell}_i \geq \alpha \mid y_i \in \mathcal{C}\right] \geq 1 - \epsilon_\ell, \quad \Pr[\tilde{g}_i \geq \alpha \mid y_i \in \mathcal{C}] \geq 1 - \epsilon_g$. Then,*

$$\Pr[V(y_i) = \texttt{accept} \mid y_i \in \mathcal{C}] \geq 1 - (\epsilon_\ell + \epsilon_g).$$

*Proof.* Both $\tilde{\ell}_i$ and $\tilde{g}_i$ independently provide high probability acceptance for correct steps. Since the ensemble score satisfies $r_i \geq \min(\tilde{\ell}_i, \tilde{g}_i)$, the probability of rejection is bounded by the union of individual error events. Then, the total error probability is at most $\epsilon_\ell + \epsilon_g$ and thus the lemma follows. $\qquad\square$

**Lemma 2** (Efficiency Guarantee). *Let $\pi_i = \Pr[V(y_i) = \texttt{accept}]$. Then the expected no. of target calls (call it $C_T$) is $\mathbb{E}[C_T] = \sum_{i=1}^T (1 - \pi_i)$. If $\pi_i \geq \pi_{\min}$ for all $i$, then $\mathbb{E}[C_T] \leq T \cdot (1 - \pi_{\min})$.*

---

**Algorithm 2:** Self-Consistency Selector

---

**Input:** Set of $k$ candidates $\{y^{(1)}, \ldots, y^{(k)}\}$, sentence transformer model $\mathcal{E}$
**Output:** Index $j^*$ of the selected candidate

1. Compute embeddings for all candidates: $e^{(j)} \leftarrow \mathcal{E}(y^{(j)})$ for $j = 1 \ldots k$
2. Normalize embeddings so that $\|e^{(j)}\|_2 = 1$
3. Compute pairwise similarity matrix: $S_{ij} \leftarrow \langle e^{(i)}, e^{(j)} \rangle$ for $i, j = 1 \ldots k$
4. Apply row-wise softmax: $\tilde{S}_{ij} = \dfrac{\exp(S_{ij})}{\sum_{l=1}^{k} \exp(S_{il})}$
5. Extract diagonal scores: $d_j \leftarrow \tilde{S}_{jj}$ for $j = 1 \ldots k$
6. $j^* \leftarrow \arg\min_j d_j$ ;     // Select most semantically consistent candidate

---

*Proof.* At each step $i$, a target call is required if and only if $V(y_i) = \texttt{reject}$. Thus, the expectation is $\sum_i (1 - \pi_i)$. If $\pi_i \geq \pi_{\min}$, the sum is bounded by $T(1 - \pi_{\min})$. This formalizes that higher acceptance rates directly reduce expected target calls. $\square$

**Theorem 1** (Accuracy–Efficiency Trade-off). *Suppose correct steps satisfy Lemma 1 and the incorrect steps are rejected with probability at least $1 - \delta$. Then, for any sequence of length $T$,*

$$\Pr[\textit{all accepted steps are correct}] \geq (1 - \epsilon_\ell - \epsilon_g)^T \cdot (1 - \delta)^{C_T}.$$

*Proof.* By Lemma 1, the probability of accepting only the correct steps is at least $(1 - \epsilon_\ell - \epsilon_g)^T$. By assumption, incorrect steps are rejected with probability at least $1 - \delta$, and there are $C_T$ target calls. Thus, the lower bound of the joint probability is $(1 - \epsilon_\ell - \epsilon_g)^T (1 - \delta)^{C_T}$. $\square$

These results show that INFERSPEC provides multiplicative accuracy guarantees while bounding the expected number of target calls.

## 4 EXPERIMENTS

Our experiments are designed to address the following research questions:
**RQ I.** Does INFERSPEC provide measurable accuracy improvements on multi-step reasoning benchmarks compared to state-of-the-art methods, while mitigating error cascades?
**RQ II.** How does the number of sampled candidates per reasoning step influence both the reliability and stability of INFERSPEC under the ensemble verification criterion?
**RQ III.** Can INFERSPEC reduce inference latency relative to reward-guided speculative decoding (RSD) while preserving, or even enhancing, accuracy guarantees?

### 4.1 EXPERIMENTAL SETUP

**Datasets and Metrics:** We conduct extensive experiments on datasets that require complex reasoning, including MATH500 Hendrycks et al. (2021), GSM8K Cobbe et al. (2021), GaoKao-2023-En Liao et al. (2024), and OlympiadBench He et al. (2024). For evaluation, we adopt the official metrics, i.e., *exact match* (EM). Detailed descriptions of the datasets can be found in Appendix A.1.

**Models:** To evaluate the effectiveness of INFERSPEC, we consider both general-purpose and math-focused LLMs as target and draft models, namely Qwen-2.5-Math Yang et al. (2024), Qwen-2.5 Qwen et al. (2025), and Llama-3 Dubey et al. (2024). For RSD, we adopt Skywork-o1-OpenPRM o1 Team (2024) as the process reward model (PRM).

**Baselines:** We evaluate INFERSPEC against four categories of baselines: (1) *Target model only*: the target model is used independently, which generally incurs a higher computational cost compared to INFERSPEC. (2) *Draft model with or without PRM:* This group covers inference time compute techniques that aim to maximize the performance of the draft model. Specifically, we evaluate majority voting and Best-of-N (BoN) Brown et al. (2024); Cobbe et al. (2021), where BoN selects the highest scoring response (last step) among N candidates using a PRM; beam search Chen et al. (2024a), which employs a PRM to identify the optimal decoding trajectory; and we process Best-of-N, which samples N candidate steps and chooses the one with the highest reward. (3) *Speculative*

Table 1: Accuracy on reasoning benchmarks.

| Method | Target Model | Draft Model | MATH500 | GSM8K | Gaokao 2023 En | Olympiad Bench |
|---|---|---|---|---|---|---|
| **Math Model, Draft and Target: Qwen2.5-Math-Instruct** | | | | | | |
| Target Model | 7B | - | 83.0 | 94.7 | 66.8 | 40.6 |
| Target-only Majority | 7B | - | 84.9 | 95.2 | 68.8 | 41.0 |
| Draft-only Majority | - | 1.5B | 79.0 | 88.9 | 67.9 | 40.9 |
| Best-of-$N$ | - | 1.5B | 82.2 | 93.3 | 67.4 | 40.7 |
| SD | 7B | 1.5B | 82.4 | 94.2 | 66.3 | 39.4 |
| RSD | 7B | 1.5B | 82.4 | 94.4 | 68.5 | 39.6 |
| RSD Majority | 7B | 1.5B | 78.0 | 88.7 | 64.9 | 38.7 |
| SC + LPBV | 7B | 1.5B | 83.2 | 94.5 | 67.5 | 39.7 |
| INFERSPEC GREEDY | 7B | 1.5B | 83.6 | 95.6 | 68.8 | 40.7 |
| INFERSPEC | 7B | 1.5B | **85.4** | **95.8** | **69.4** | **41.2** |
| **General Model, Draft and Target: Qwen2.5-Instruct** | | | | | | |
| Target Model | 7B | - | 74.8 | 91.7 | 64.9 | 38.8 |
| Draft-only Majority | - | 1.5B | 66.4 | 82.1 | 56.9 | 28.7 |
| Best-of-$N$ | - | 1.5B | 73.4 | 89.7 | 60.5 | 32.7 |
| SD | 7B | 1.5B | 74.8 | 91.6 | 63.1 | 37.1 |
| RSD | 7B | 1.5B | 71.4 | 90.1 | 60.5 | 37.6 |
| RSD Majority | 7B | 1.5B | 60.6 | 77.0 | 55.3 | 31.7 |
| INFERSPEC GREEDY | 7B | 1.5B | 74.9 | 92.0 | 65.5 | 37.8 |
| INFERSPEC | 7B | 1.5B | **77.0** | **93.0** | **66.0** | **40.3** |
| **General Model, Draft: Llama-3.2-Instruct and Target: Llama-3.1-Instruct** | | | | | | |
| Target Model | 8B | - | 48.2 | 83.9 | 40.8 | 14.5 |
| Draft-only Majority | - | 1B | 38.0 | 60.2 | 32.2 | 9.5 |
| Best-of-$N$ | - | 1B | 48.6 | 74.8 | 40.7 | 14.4 |
| SD | 8B | 1B | 47.0 | 83.4 | 40.1 | 16.1 |
| RSD | 8B | 1B | 50.0 | 83.9 | 41.8 | 15.7 |
| RSD Majority | 8B | 1B | 36.6 | 61.9 | 30.6 | 12.3 |
| INFERSPEC GREEDY | 8B | 1B | 50.0 | 84.5 | 41.9 | 16.9 |
| INFERSPEC | 8B | 1B | **51.6** | **85.1** | **43.9** | **17.2** |

*decoding (SD):* We also include speculative decoding with 7 speculative tokens, a technique aimed at accelerating inference Leviathan et al. (2023). (4) *RSD:* Liao et al. (2025) leverages a PRM to score intermediate steps and adaptively determine when to call the target model.

**Setting:** We perform all experiments on NVIDIA A100 GPUs with vLLM as the serving backend. We define a reasoning step as a generation terminated by \n\n. For generating multiple samples, we set $temperature = 0.7$, $top\_p = 0.8$, and $n = 16$. INFERSPEC refers to this multi-sample setting combined with our self-consistency selector, which chooses the most representative reasoning step. RSD Majority likewise employs this multi-sample setting, with each step scored by a PRM and the highest-scoring candidate selected. In the greedy setting where $temperature = 0$, $top\_p = 1$, and $n = 1$, we refer to our approach as INFERSPEC GREEDY. For both RSD and INFERSPEC, we set the threshold parameter to $\tau = 0.7$ and we set $\beta = 0.3$ for INFERSPEC. For details about hyperparameters refer to Appendix A.2.2. Unless stated otherwise, all models used are Qwen-2.5-Math-Instruct.

## 4.2 PERFORMANCE COMPARISON

To address **RQ1**, we evaluate INFERSPEC on a broad set of challenging reasoning benchmarks, as summarized in Table 1, and make the following observations: (1) Inference-time compute strategies such as majority voting and Best-of-N, which rely on extensive draft sampling, typically underperform compared to the accuracy of a single target model. This underscores the critical role of larger models in reasoning tasks, as their capabilities cannot be readily matched by smaller models even with increased computation. (2) While target-only majority voting may match INFERSPEC in accuracy, it incurs substantially higher computational cost as every reasoning step must be sampled multiple times from the target, contrary to our objective of reducing target calls. (3) Although speculative decoding (SD) is theoretically unbiased, ensuring accuracy equivalent to the target model,

Table 2: Comparison with search-based methods on Qwen2.5-Math-Instruct. Beam Search and Process Best-of-$N$ use a 1.5B base model and a 1.5B PRM.

| Method | Setting | MATH500 | GSM8K |
|---|---|---|---|
| Draft Model (1.5B) | - | 73.8 | 85.0 |
| Process Best-of-$N$ | $N = 8$ | 75.8 | 87.8 |
| Process Best-of-$N$ | $N = 16$ | 76.0 | 87.9 |
| Beam Search | bs=4 | 78.2 | 88.9 |
| Beam Search | bs=8 | 78.2 | 88.4 |
| RSD (1.5B/7B/1.5B) | - | 82.4 | 94.4 |
| INFERSPEC GREEDY | - | 83.6 | 95.6 |
| INFERSPEC | maj@16 | **85.4** | **95.8** |

it often performs worse in practice. As also reported by Chen et al. (2023), this drop arises from floating-point errors. Furthermore, when the draft model surpasses the target model, the strict un-biased nature of SD can actually degrade performance relative to the draft. Hence, deploying SD requires careful consideration of such cases. (4) Reward-guided speculative decoding (RSD) alleviates this limitation by incorporating a process reward model (PRM) to assess the quality of draft reasoning steps. However, relying on an external verifier introduces both latency and computational overhead. (5) INFERSPEC replaces PRMs with lightweight model-internal grounded verifiers to evaluate the draft steps. Across all benchmarks, INFERSPEC consistently exceeds both the single target model and RSD, demonstrating the strength and efficiency of our approach. In Appendix A.2.3, we show **qualitative analysis** of reasoning steps scored by PRM. Even though all draft-generated steps receive high acceptance scores from the PRM, the final answer is still incorrect, highlighting the *need for stronger verification methods that ensure both step-wise soundness and final-answer correctness.* While LPBV captures confidence, it lacks grounding, so confident but ungrounded steps frequently slip through. In contrast, INFERSPEC achieves higher accuracy, demonstrating that ABGV is essential for rejecting ungrounded steps that appear locally plausible.

### 4.3 COMPARISON WITH SEARCH-BASED APPROACHES

We also compare INFERSPEC with beam search Chen et al. (2024a) and process Best-of-N, as reported in Table 2. Our method consistently outperforms both search-based baselines. These findings reveal an important observation: When reasoning steps become particularly complex, search-based techniques face limitations, as combinatorial growth of candidate solutions makes it difficult to reliably identify optimal paths, resulting in degraded performance. In contrast, INFERSPEC leverages the expressive power of larger models to generate strong candidate solutions. In addition, the incorporation of an ensemble verifier provides step-level feedback, mitigating the challenges of difficult reasoning tasks. *This highlights that moving beyond purely search-based strategies and augmenting*

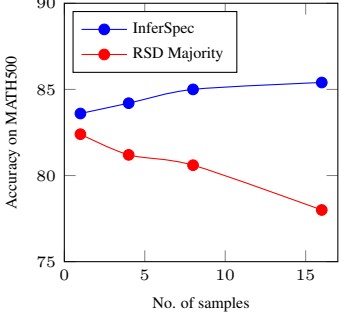

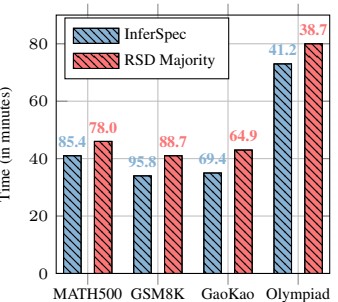

(a) Varying number of samples

(b) Runtime comparison

Figure 2: (a) Varying number of samples (b) Runtime comparison (y-axis) RSD Majority vs INFER-SPEC with corresponding **accuracy indicated on top of bars**.

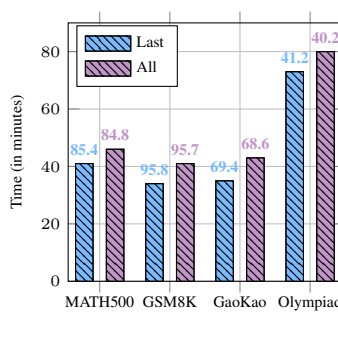 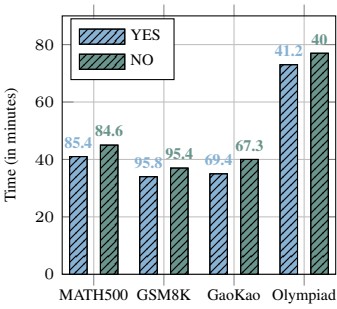

(a) Changing Layers            (b) Sparsity in attention heads

Figure 3: Ablation studies: (a) changing layers (b) Sparsity in attention heads by INFERSPEC. Runtime comparison (y-axis) with corresponding **accuracy indicated on top of bars**.

*larger models with lightweight feedback mechanisms can deliver both higher efficiency and stronger performance, particularly when the search space is vast or the reasoning task is highly complex.*

### 4.4 EFFECT OF SAMPLE SIZE AND RUNTIME COMPARISON

To investigate **RQ II**, we evaluate INFERSPEC under varying sample sizes per reasoning step. As shown in Figure 2(a), accuracy steadily improves as more diverse candidates are explored, with gains saturating at higher counts. In contrast, RSD Majority exhibits diminishing returns and even degrades performance with larger samples, due to accumulated noise from PRM. These findings demonstrate that INFERSPEC takes advantage of additional candidate generations more effectively.

For **RQ III**, Figure 2(b) compares the runtime of RSD Majority and INFERSPEC, with the accuracy indicated on top of the bars. INFERSPEC consistently achieves both higher accuracy and lower latency. For example, on GSM8K, INFERSPEC achieves an accuracy of $95.8\%$ in 34 minutes, compared to the accuracy of the RSD Majority $88.7\%$ in more than 41 minutes. Together, these results confirm that our ensemble verifier-guided speculative decoding framework improves reasoning reliability while delivering superior efficiency.

### 4.5 ABLATION STUDIES

We perform ablation studies to examine key design choices in INFERSPEC, focusing on (a) the layers used to extract internal signals and (b) role of sparsity in attention heads, as shown in Figure 3.
**Changing Layers:** Figure 3(a) compares using attention from the last three layers versus aggregating across all layers. Although the latter yields marginal gains on some benchmarks (e.g. GSM8K), it consistently adds runtime overhead. Leveraging only the last three layers strikes a better balance, achieving higher accuracy with lower latency. We show other variants in Appendix A.2.1.
**Sparsity in Attention Heads:** We discard entries below 0.01 in attention heads. Figure 3(b) shows that enforcing sparsity improves both accuracy and runtime. This suggests that sparsity sharpens the focus of the verifier on relevant attention patterns, enhancing efficiency without loss of performance.

### 5 CONCLUSION AND FUTURE WORK

In this work, we propose INFERSPEC, an adaptive speculative decoding that improves both efficiency and accuracy in multistep reasoning. By leveraging lightweight model-internal signals for verification, along with a self-consistency selector that identifies semantically representative reasoning step across samples, INFERSPEC avoids dependence on external reward models and achieves higher accuracy with reduced latency compared to state-of-the-art methods. In future, we plan to extend INFERSPEC by incorporating additional internal signals such as entropy-based measures and uncertainty calibration to refine verifier reliability. Another promising direction is applying INFER-SPEC to domains beyond text reasoning, including code generation and multimodal tasks.

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

# A APPENDIX

## A.1 DATASETS DESCRIPTION

An overview of the dataset statistics and examples are shown in Table 3.

Table 3: Overview of the Complex QA datasets used in this study.

| Dataset | #Test | Example Question | Description |
|---------|-------|------------------|-------------|
| **MATH500** Hendrycks et al. (2021) | 500 | What is the smallest positive perfect cube that can be written as the sum of three consecutive integers? | multi-step arithmetic word problems |
| **GSM8K** Cobbe et al. (2021) | 1319 | The red car is 40% cheaper than the blue car. The price of the blue car is $100. How much do both cars cost? | multi-step arithmetic word problems |
| **GaoKao-2023-En** Liao et al. (2024) | 385 | Suppose the universe set is U={0,1,2,4,6,8}. Two of its subsets are M={0,4,6}, N={0,1,6}. Find $M \cup \bar{N}$. | multi-step arithmetic word problems |
| **OlympiadBench** He et al. (2024) | 675 | A number is called Norwegian if it has three distinct positive divisors whose sum is equal to 2022. Determine smallest Norwegian number. | multi-step arithmetic word problems |

**MATH500**: A benchmark subset curated from the MATH dataset, consisting of 500 competition-level mathematics problems spanning algebra, geometry, combinatorics, number theory, and probability. Each problem is accompanied by a detailed step-by-step solution, requiring multi-hop symbolic and logical reasoning. We use the full 500 problems as the evaluation set.

**GSM8K**: A dataset of linguistically diverse grade-school math word problems designed to test multi-step numerical reasoning. It comprises 8.5K questions, with a test set of 1,319 problems. Each question includes annotated solutions with intermediate steps, encouraging models to demonstrate faithful reasoning chains.

**Gaokao-2023-En**: Derived from the English-translated 2023 Gaokao (China's national college entrance exam), this dataset contains high-school level math word problems with a strong emphasis on reasoning over algebra, functions, and applied mathematics. It poses particular challenges due to its formal problem style and complex solution trajectories. The evaluation set includes 385 problems.

**OlympiadBench**: A large-scale benchmark of problems drawn from global mathematics and science Olympiads, covering topics such as advanced algebra, geometry, physics, and logical reasoning. The problems are highly challenging, requiring creative multi-step reasoning far beyond routine computation. We evaluate on the test split of 675 questions.

## A.2 ADDITIONAL EXPERIMENTS

### A.2.1 CHANGING LAYERS

We perform ablation studies to analyze key design choices in INFERSPEC, particularly the selection of layers used to extract internal grounding signals. Figure 4 compares four settings: using attention from (1) all layers, (2) the last three layers, (3) the middle three layers, and (4) the first three layers. The results show that the first three layers perform noticeably worse than other variants, while the middle three layers achieve higher accuracy but still lag behind the deeper layers. Both the last three layers and all layers yield strong performance, but using all layers incurs higher runtime overhead. Overall, the last three layers provide the best trade-off, delivering strong accuracy with lower latency.

### A.2.2 TUNING OF $\beta$ AND $\tau$

We analyze the sensitivity of our approach to two hyperparameters: the step acceptance threshold $\tau$ for our ensemble verifier, and the weighting factor $\beta$, which balances the log-probability and attention grounding score when computing the ensemble score. As shown in Table 4, accuracy remains stable across different values of $\beta$, with $\beta = 3$ performing slightly better than higher values, suggesting that a moderate weighting strikes a good balance between model confidence and grounding. Table 5 reports results for varying $\tau$. We find that $\tau = 0.7$ achieves the most consistent gains across datasets, while both lower ($\tau = 0.6$) and higher values ($\tau = 0.8, 0.9$) lead to small drops. Over-

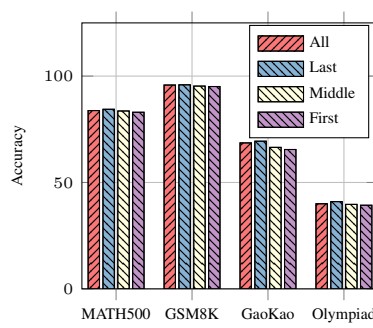

Figure 4: Changing Layers

Table 4: Accuracy with different $\beta$s. Overall, $\beta = 0.3$ works well for different tasks.

| Method | Target Model | Draft Model | Setting | MATH500 | GSM8K | Gaokao 2023 En | Olympiad Bench |
|---|---|---|---|---|---|---|---|
| Math Model, Draft and Target: Qwen2.5-Math-Instruct | | | | | | | |
| Ours Majority | 7B | 1.5B | $\beta = 0.3$ | 85.4 | 95.8 | 69.4 | 41.2 |
| Ours Majority | 7B | 1.5B | $\beta = 0.5$ | 85.0 | 95.7 | 68.5 | 40.4 |
| Ours Majority | 7B | 1.5B | $\beta = 0.7$ | 84.4 | 95.6 | 65.5 | 40.2 |

Table 5: Accuracy with different $\tau$s. Overall, $\tau = 0.7$ works well for different tasks.

| Method | Target Model | Draft Model | Setting | MATH500 | GSM8K | Gaokao 2023 En | Olympiad Bench |
|---|---|---|---|---|---|---|---|
| Math Model, Draft and Target: Qwen2.5-Math-Instruct | | | | | | | |
| Ours Majority | 7B | 1.5B | $\tau = 0.6$ | 83.6 | 93.5 | 67.4 | 39.7 |
| Ours Majority | 7B | 1.5B | $\tau = 0.7$ | 85.4 | 95.8 | 69.4 | 41.2 |
| Ours Majority | 7B | 1.5B | $\tau = 0.8$ | 84.2 | 94.6 | 68.7 | 40.4 |
| Ours Majority | 7B | 1.5B | $\tau = 0.9$ | 85.1 | 95.6 | 69.2 | 41.0 |

all, our method is robust to hyperparameter choices, with $\beta = 3$ and $\tau = 0.7$ serving as effective defaults across tasks.

### A.2.3 QUALITATIVE ANALYSIS

Table 6, 7 presents a qualitative example of reasoning steps scored by the PRM. Each intermediate step receives a high score, leading the verifier to accept the draft-generated reasoning without intervention. However, despite this consistent acceptance, the reasoning chain ultimately produces an incorrect final answer. This illustrates a key limitation of relying solely on PRM scores: while they may capture local plausibility of individual steps, they do not guarantee global correctness of the overall solution. Such cases highlight the need for more robust evaluation mechanisms that can account for consistency across steps as well as correctness of the final outcome.

### A.3 COMPLEXITY ANALYSIS - INFERSPEC

We compare the computational complexity of Speculative Decoding (SD), Reward-guided Speculative Decoding (RSD), and INFERSPEC.

Let:

- $T$ = number of reasoning steps,
- $k$ = number of draft candidates sampled per step,
- $d$ = hidden dimension of embeddings,
- $H$ = number of attention heads,

Table 6: Qualitative analysis of reasoning steps scored by PRM. Even though all draft-generated steps receive high acceptance scores from the PRM, the final answer is still incorrect.

*Question: A treasure hunter found a buried treasure chest filled with gems. There were 175 diamonds, 35 fewer rubies than diamonds, and twice the number of emeralds than the rubies. How many of the gems were there in the chest?*

| Reasoning Steps | PRM Scores |
|---|---|
| Step 1: Identify the number of diamonds in the chest. The treasure hunter found 175 diamonds. | 0.962108 |
| Step 2: Calculate the number of rubies. There were 35 fewer rubies than diamonds, so the number of rubies is 175 - 35 = 140. | 0.962108 |
| Step 3: Calculate the number of emeralds. The number of emeralds is twice the number of rubies, so the number of emeralds is 2 * 140 = 280. | 0.962108 |
| Step 4: Calculate total number of gems. To find the total number of gems, add the number of diamonds, rubies, and emeralds: 175 + 140 + 280 = 535. | 0.961534 |
| Step 5: The final answer is: \boxed{535} | 0.962108 |

Table 7: Qualitative analysis of reasoning steps scored by PRM. Even though all draft-generated steps receive high acceptance scores from the PRM, the final answer is still incorrect.

*Question: Two girls each got 1/6 of the 24 liters of water. Then a boy got 6 liters of water. How many liters of water were left?*

| Reasoning Steps | PRM Scores |
|---|---|
| Step 1: Calculate the amount of water each girl received. Each girl got 1/6 of 24 liters of water, which is (1/6) * 24 = 4 liters. | 0.92192 |
| Step 2: Calculate the total amount of water the girls received. Since there are two girls, the total amount of water they received is 2 * 4 = 8 liters. | 0.92079 |
| Step 3: Calculate the amount of water left after the boy got 6 liters. Subtract the amount of water the boys received from the total amount of water the girls received: 8 - 6 = 2 liters. | 0.92079 |
| Step 4: The final answer is: \boxed{2} | 0.92079 |

- $L$ = number of layers used in ABGV,
- $C_{\text{draft}}, C_{\text{target}}$ = per-step cost of draft and target models.

**Speculative Decoding (SD).**

$$\text{Complexity} = O(T \cdot (k \cdot C_{\text{draft}} + (1 - \pi) \cdot k \cdot C_{\text{target}})), \tag{1}$$

where $\pi$ is the acceptance probability.

**Reward-guided Speculative Decoding (RSD).** In addition to SD cost, RSD requires a pretrained reward model (PRM) verifier:

$$\text{Complexity} = O(T \cdot (k \cdot C_{\text{draft}} + (1 - \pi) \cdot k \cdot C_{\text{target}} + C_{\text{PRM}})). \tag{2}$$

**INFERSPEC.** It includes draft sampling, self-consistency selector, and ensemble verification.

$$\text{Complexity} = O\left(T \cdot \left(k \cdot C_{\text{draft}} + k^2 d + \tilde{L}\tilde{H}T^2 + (1 - \pi) \cdot k \cdot C_{\text{target}}\right)\right), \tag{3}$$

where $\tilde{L} \ll L$ and $\tilde{H} \ll H$ represent the reduced number of layers and heads used in ABGV under sparsity/last-layer approximations.

It is not difficult to infer the following from the complexity analysis.

- INFERSPEC avoids the external PRM cost $C_{\text{PRM}}$ in RSD, reducing latency and compute.

Table 8: Qualitative analysis of reasoning steps accepted by INFERSPEC, where consistent step verification leads to the correct final answer.

| |
|---|
| *Question: A treasure hunter found a buried treasure chest filled with gems. There were 175 diamonds, 35 fewer rubies than diamonds, and twice the number of emeralds than the rubies. How many of the gems were there in the chest?* |
| **Reasoning Steps** |
| Step 1: First, let's identify the number of diamonds, which is given as 175. |
| Step 2: The number of rubies is 35 fewer than the number of diamonds, so we need to subtract 35 from 175 to find the number of rubies: 175 - 35 = 140 rubies. |
| Step 3: The number of emeralds is twice the number of rubies, so we need to multiply the number of rubies by 2 to find the number of emeralds: 140 * 2 = 280 emeralds. |
| Step 4: To find the total number of gems in the chest, we need to add the number of diamonds, rubies, and emeralds: 175 + 140 + 280 = 595 gems. |
| Step 5: The final answer is: \boxed{595} |

Table 9: Qualitative analysis of reasoning steps accepted by INFERSPEC, where consistent step verification leads to the correct final answer.

| |
|---|
| *Question: Two girls each got 1/6 of the 24 liters of water. Then a boy got 6 liters of water. How many liters of water were left?* |
| **Reasoning Steps** |
| Step 1: First, let's calculate the amount of water each girl got. Since they each got 1/6 of the 24 liters, we can calculate this as 24 * (1/6) = 4 liters per girl. |
| Step 2: Since there are two girls, the total amount of water they got is 4 * 2 = 8 liters. |
| Step 3: Now, let's calculate the amount of water left after the girls got their share. We subtract the amount they got from the total amount of water: 24 - 8 = 16 liters. |
| Step 4: Then, boy got 6 liters of water. We subtract this from remaining water: 16 - 6 = 10 liters. |
| Step 5: The final answer is: \boxed{10} |

- With practical optimizations ($\tilde{L} \approx 3$, $\tilde{H} \ll H$), the ABGV overhead is negligible relative to the forward draft / target passes.

- Empirically, INFERSPEC achieves up to $11\%$ lower runtime than RSD while improving accuracy by 1–3% in benchmarks.

## A.4 LLM USAGE

Large Language Models (LLMs) were used in this work solely as general-purpose assistive tools. Specifically, they were employed in two limited capacities: (i) to aid in polishing the writing for clarity and readability, and (ii) to assist in retrieval and discovery tasks, such as identifying related work. No part of the research design, algorithm development, theoretical analysis, or experimental implementation relied on LLMs. Their role was restricted to supportive tasks.

