# OpenReview forum: "InferSpec: Adaptive Inference-Time Compute with Ensemble Verifier-Guided Speculative Decoding for Efficient Reasoning"
_ICLR.cc/2026/Conference — ICLR 2026 Conference Withdrawn Submission_

### Official Review · Reviewer_v1Ds · 2025-10-30

**Soundness:** 2
**Presentation:** 3
**Contribution:** 2
**Rating:** 4
**Confidence:** 4

**Summary:**

This paper proposes INFERSPEC, a speculative decoding framework for multi-step reasoning that aims to replace expensive external reward models. It uses a lightweight "ensemble verifier" based on two model-internal signals: Attention-Based Grounding Verification (ABGV) and Log-Probability-Based Verification (LPBV). This verifier adaptively accepts draft steps or escalates to a stronger target model.

**Strengths:**

1.	Important Problem: The paper tackles the critical problem of high inference cost in LLM reasoning.

2.	Clear Motivation: The goal of replacing heavy, external Process Reward Models (PRMs) with lightweight, model-internal signals is excellent and practical.

**Weaknesses:**

1.	(Major Weakness) Lack of Foundational Validation: The paper fails to provide any direct evidence (e.g., correlation studies) that its two core signals, ABGV and LPBV, are meaningful proxies for step-wise reasoning correctness. The entire proposal is built on an unproven premise.

2.	Lack of Generality: The paper's claims of efficiency and accuracy are only supported by experiments on mathematical and formal reasoning datasets (MATH500, GSM8K, etc.) . It is completely unclear if this method has any utility in open-ended or creative generative tasks.

3.	Missing Critical Ablation: The method combines the new verifier (ABGV+LPBV) with a Self-Consistency Selector. The paper does not ablate the effect of the selector, making it impossible to know if the small accuracy gains come from the novel verifier or just from applying the known self-consistency technique.

**Questions:**

1.	Can the authors provide a direct quantitative analysis (e.g., a correlation) to demonstrate that the proposed ABGV and LPBV scores actually correlate with ground-truth step-wise correctness?

2.	What is the performance of a baseline that uses only the Self-Consistency Selector to choose from k draft candidates, but uses a simpler verification rule (e.g., standard SD or just LPBV)? This is critical to isolate the true contribution of the ABGV verifier.

---

> ### Author Response · Authors · 2025-11-26
> **Point-by-point responses to Reviewer v1Ds**
>
> Dear Reviewer v1Ds,
>
> Thank you very much for providing us with valuable feedback. We appreciate the detailed comments. Below, we have provided responses to each of your comments.
>
> ***Quantitative analysis to demonstrate that the proposed ensemble (ABGV and LPBV) score correlates with ground-truth step-wise correctness***
> > We would like to clarify that we have **already included** a quantitative step-wise analysis in the Appendix (Tables 6–9) demonstrating how our ensemble (ABGV and LPBV) score relates to ground-truth correctness.
> > - *Tables 6 and 7* show cases where a Process Reward Model *(PRM) assigns uniformly high scores to incorrect reasoning steps*, illustrating that PRM-based validation lacks correlation with actual correctness.
> > - *Tables 8 and 9*, in contrast, show that our ensemble (ABGV and LPBV) scores successfully differentiate correct from incorrect steps. The steps with poor grounding or low log-probability are rejected, while **well-grounded, high-confidence steps are accepted**, resulting in correct final answers.
> > This step-level quantitative evidence demonstrates that our internal verification signals track correctness much more reliably than PRM signals. Our ensemble (ABGV and LPBV) scores exhibit discriminative correlation, as reflected by their consistent rejection of faulty steps and acceptance of correct ones.
>
> ***Performance of baseline that uses Self-Consistency Selector to choose from k draft candidates, but uses a simpler verification rule (e.g., LPBV)?***
> >  We provide results for SC + LPBV in the table below. While LPBV captures confidence, it lacks grounding, so confident but ungrounded steps frequently slip through. In contrast, InferSpec (SC + Ensemble) achieves higher accuracy compared to SC + LPBV, demonstrating that **ABGV is essential for rejecting ungrounded or spurious reasoning steps** that appear locally plausible but are not connected to the problem context. We have included and discussed these baseline results in **Section 4.2, lines 402-404**, in the updated manuscript to make ABGV’s contribution explicit.
> > |**Method**        |  MATH500 | GSM8K   | Gaokao  |  Olympiad  |
> > |---------------------|----------------|--------------|-------------|----------------|
> > | SC + LPBV         |  83.2          |  94.5        |   67.5      |  39.7          |
> > | **InferSpec**           |  **85.4**          |  **95.8**        |   **69.4**      |  **41.2**          |
>
> We would like to express once again our gratitude to Reviewer v1Ds for their valuable comments and suggestions. We believe our responses above effectively address all of Reviewer v1Ds's concerns and further enhance the quality of our work.
>
> Sincerely,
>
> The Authors

---

### Official Review · Reviewer_GsNE · 2025-11-01

**Soundness:** 3
**Presentation:** 2
**Contribution:** 2
**Rating:** 4
**Confidence:** 4

**Summary:**

This paper proposes InferSpec, a framework that enhances speculative decoding for LLM reasoning tasks through adaptive inference-time compute allocation. Different from Standard speculative decoding (SD) and reward-guided SD (RSD), InferSpec replaces external verifiers with two model-internal verification signals: (i) Attention-Based Grounding Verification (ABGV), which uses attention rollout matrices to ensure outputs are grounded in inputs or validated prior steps, and (ii) Log-Probability-Based Verification (LPBV), which enforces token-level confidence. At each reasoning step, InferSpec samples multiple draft candidates, applies a self-consistency selector to identify the most representative one, then evaluates it using a weighted ensemble of ABGV and LPBV scores. Steps passing the threshold are accepted; others trigger target model recomputation. Experiments on MATH500, GSM8K, Gaokao-2023-En, and OlympiadBench show InferSpec improves accuracy by up to 3.6% while reducing latency by ~11% compared to RSD and standard SD.

**Strengths:**

1. InferSpec introduces complementary internal signals (ABGV via attention rollout, LPBV via log-probabilities) that eliminate external PRM overhead while maintaining generalizability.
2. Thorough evaluation across four reasoning benchmarks, three model families, and multiple baselines (majority voting, Best-of-N, beam search, SD, RSD). Systematic ablations examine design choices, and qualitative analysis effectively illustrates PRM failure modes.
3. InferSpec consistently achieves higher accuracy with lower latency than RSD (e.g., 95.8% in 34 minutes vs. 88.7% in 41 minutes on GSM8K). Complexity analysis confirms practical efficiency gains from avoiding external verifiers.

**Weaknesses:**

1. The approach exhibit limited domain generalizability. All experiments focus exclusively on mathematical reasoning. No evidence demonstrates that ABGV and LPBV generalize to code generation, commonsense reasoning, or other domains where attention patterns and confidence distributions may differ substantially.
2. The parameter setting of InferSpec seems to be crucial and determined empirically. Self-consistency selector overhead are not compared against simpler strategies.
3. Authors should consider comparing the InferSpec to more recent baselines such as SpecReason: https://arxiv.org/abs/2504.07891 by Rui Pan, Yinwei Dai, Zhihao Zhang, etc.

**Questions:**

1. Have authors evaluated InferSpec on non-mathematical reasoning domains (e.g., code generation, commonsense reasoning, multi-hop QA) to validate whether ABGV and LPBV generalize beyond math-specific attention and confidence patterns?
2. ow does InferSpec compare against more recent speculative reasoning methods such as SpecReason (Pan et al., 2025)?

---

> ### Author Response · Authors · 2025-11-26
> **Point-by-point responses to Reviewer GsNE**
>
> Dear Reviewer GsNE,
>
> Thank you very much for providing us with valuable feedback. We appreciate the detailed comments. Below, we have provided responses to each of your comments.
>
> ***InferSpec on non-mathematical reasoning domains***
> > Thank you for the question. To evaluate whether InferSpec generalizes beyond mathematical reasoning, we additionally tested it on a **non-mathematical task (NL2SQL)**.
> > We provide results for NL2SQL task in the table below. InferSpec continues to outperform both *Target-only* and *RSD* baselines, showing consistent gains in step correctness and final accuracy. These results indicate that InferSpec **generalize well to non-math domains**.
> > |**Method**        |  NL2SQL   |
> > |---------------------|----------------|
> > | Draft-only          |       62.50   |
> > | Taget-only         |       67.18    |
> > | RSD                 |       68.52   |
> > | **InferSpec**    |  **73.43**    |
>
>
>
> ***How does InferSpec compare against SpecReason?***
> > - SpecReason [1] performs step-level speculative reasoning using a lightweight speculator, while the **base target model acts as a critic** to evaluate the **semantic utility** of each proposed step. It uses a single-token utility score with a threshold-based acceptance rule where accepted steps are retained, and rejected steps are regenerated by the base target model.
> > - **In contrast, InferSpec** couples **multi-sample self-consistency** with an **ensemble verifier** (ABGV for grounding and LPBV for confidence), enabling **stronger rejection of ungrounded steps** and offering **formal soundness and efficiency guarantees**. While SpecReason primarily focuses on semantic plausibility, InferSpec is designed to improve **robustness and reliability**, particularly against *plausible but ungrounded intermediate steps*. We have cited and discussed SpecReason in **Section 2, lines 93-95** (marked in blue in the revised manuscript).
>
> > [1] SpecReason: Fast and Accurate Inference-Time Compute via Speculative Reasoning, Pan et al. NeurIPS 2025.
>
> We would like to thank Reviewer GsNE for the valuable comments. We believe that our responses above effectively address all of Reviewer GsNE 's concerns and contribute to further strengthening our work.
>
> Sincerely,
>
> The Authors

---

### Official Review · Reviewer_adcc · 2025-11-01

**Soundness:** 3
**Presentation:** 3
**Contribution:** 3
**Rating:** 4
**Confidence:** 4

**Summary:**

This paper focuses on efficient generation by incorporating SLM and LLM. Traditional speculative decoding (SD) method requires a strict distribution alighnment between LLM and SLM, and its token-centric nature admits subtle flaws in intermediate steps to propagate. A recent variant, RSD, relies on external reward, introducing latency.

In this paper, the authors propose InferSpec that samples multiple steps with SLM, selects a reprentative one with semantic similarity, then evaluates it with two model-internal criteria (attention-based grounding verification and log-probability based verification). If the selected step is rejected, the LLM will sample multple steps, chooses the representative one. And the decoding is switch again to SLM.

With theoretical justification and extensive experiments, InferSpec shows comparable or better accuracy and efficiency than SD and RSD. Detailed ablation studies also justify the dsign choice.

**Strengths:**

1. The paper is well-written with a clear motivation, i.e. using model-internal signal to verify the step instead of external reward.
2. The theoretical justification supports the proposed method.
3. Extensive experiments show the accuray and efficiency benefits from InferSpec.

**Weaknesses:**

1. Lack of baseline. Since InferSpec applies the target model to sample multiple steps, it's suggested to include the majority vorting results from the target model only.
2. Unfair runtime comparison. According to Table 1, it seems RSD is originally designed without majority voting. With majority voting, RSD performs much worse. When comparing runtime with RSD in Figure 2, I don't see the reason to compare with RSD majority.
3. Potential more memory consumption. According to L147, we need to save the attention matrices from each layer to compute R. This saving becomes more memory-consumed for larger model and longer output.

**Questions:**

1. Why do you still use the target model to generate multiple steps? What is the difference bwteen generating only one step and multiple steps with the target model?

### Suggestion
1. Table 1: highlight the best results for easy observation.

---

> ### Author Response · Authors · 2025-11-26
> **Point-by-point responses to Reviewer adcc**
>
> Dear Reviewer adcc,
>
> Thank you very much for providing us with valuable feedback. We appreciate the detailed comments. Below, we have provided responses to each of your comments.
>
> ***Majority voting results of target-only model***
> > Thank you for the suggestion. We agree that a **target-only majority voting** baseline is a meaningful comparison, since the target model is strong and can benefit from multiple samples.
> > - We would like to emphasize that *this baseline is orthogonal to our contribution*. **InferSpec** focuses on **reducing calls to the target model** through step-wise speculative reasoning. Therefore, even though target-only majority voting achieves similar accuracy in some settings, it typically requires **significantly higher computational cost**, since every reasoning step must be generated multiple times by the target.
> > - In contrast, InferSpec achieves competitive performance **with fewer target-model queries**, demonstrating a better **efficiency–performance trade-off**, which is the primary motivation of our work. We have made this distinction clearer in **Section 4.2, lines 375-377** (marked in blue in the revised manuscript).
> > - |       **Method**     |  MATH500  | GSM8K    | Gaokao  |  Olympiad   |
> > |-------------------------|-----------------|--------------|--------------|----------------|
> > | Target-only Majority |   84.9          |   95.2       |   68.8      |      41.0       |
> > |  **InferSpec**          |  **85.4**       |   **95.8**  |   **69.4** |    **41.2**    |
>
>
> ***Why is the runtime compared with RSD Majority in Figure 2?***
> > We appreciate the reviewer’s point. Our goal in including **RSD majority** was not to imply that it is the original design, but rather to provide a **fair apples-to-apples comparison** in the **self-consistency setting**. The runtime comparison is only meaningful when all methods operate under the same setting.
> > Additionally, we provide **runtime results for the original RSD** and **InferSpec Greedy** in the table below for completeness.
> > |**Method**          |  MATH500 | GSM8K    | Gaokao  |  Olympiad  |
> > |-----------------------|-----------------|---------------|-------------|---------------|
> > |        RSD              |     29           |        23      |       24     |     46         |
> > |  **InferSpec Greedy** |     **18**           |        **13**      |       **13**     |     **30**         |
>
> ***The saving of the attention matrices from each layer to compute R becomes more memory-consumed for larger model and longer output.***
> > Thank you for asking the question. We agree that naively storing all attention matrices could increase memory usage. To address this, our method **does not** retain attention matrices from all layers:
> > - We store attention matrices from **only the last 3 layers**, which we show (Figure 3) to be sufficient for grounding quality.
> > - We apply **sparsification in attention heads**, discarding attention entries below a **0.01 threshold**, significantly reducing memory footprint with negligible impact on performance.
> > Thus, the actual memory overhead remains low even for longer outputs. We have clarified these implementation details in **Section 3.1, lines 162-169**  (marked in blue in the revised manuscript), to avoid misunderstanding.
>
> ***What is the difference between generating only one step and multiple steps with the target model?***
> > Thank you for the question. We use the target model to generate multiple reasoning steps (rather than a single one) **to reduce stochastic variance and improve step-wise reliability**. Since even the target model *can occasionally produce locally inconsistent or semantically divergent reasoning*, relying on a single sample may propagate errors.
> > Empirically, we observe that *multi-step target sampling leads to higher final accuracy* and more stable reasoning trajectories, demonstrating that self-consistency remains beneficial even at the target model stage. We have updated **Section 3.3, lines 209-211** (marked in blue in the revised manuscript) to clarify this design choice.
>
> ***Table 1: highlight the best results for easy observation.***
> > Thank you for the suggestion. We have already highlighted the best-performing row in yellow in Table 1, and have additionally made the best numbers bold in the revised version to further improve readability and ease of comparison.
>
> We would like to express once again our gratitude to Reviewer adcc for their valuable comments and suggestions. We believe our responses above effectively address all of Reviewer adcc 's concerns and further enhance the quality of our work.
>
> Sincerely,
>
> The Authors

---

### Author Response · Authors · 2025-12-03
**Summary of Rebuttal and Revised Manuscript**

Dear ACs and Senior ACs,

We are pleased to inform you that we got insightful feedback from all the reviewers. All reviewers appreciated the work and found the work to be relevant. We have answered all their concerns and incorporated all necessary changes in the revised manuscript. Below is a summary of our discussions with each reviewer.

# Reviewer adcc:

The reviewer acknowledged several strengths of our work, including the clear motivation for leveraging model-internal signals for step verification instead of relying on external rewards, as well as the strong theoretical justification supporting the proposed method. The reviewer requested clarifications and a few new results as detailed below.

- **New Results Asked:** Majority voting results of target-only model
> New results and further clarifications are provided (Line-375-377 in Revised Manuscript).

- **Clarification Asked:** Memory consumption by attention matrices from each layer.
> Clarification is provided (Line-162-169 in Revised Manuscript).

- **Clarification Asked:** Generate only one step or multiple steps with the target model?
> Clarification is provided (Line-209-211 in Revised Manuscript).

# Reviewer GsNE:

The reviewer highlighted that our method is technically sound, eliminating external PRM overhead while maintaining generalizability, and demonstrating strong empirical performance across datasets. The ablations thoroughly examine key design choices, and the qualitative analysis clearly illustrates failure modes of PRM-based verification. The reviewer requested minor revisions and a few new results as detailed below.

- **Suggested Minor Revisions:** Citing the SpecReason paper.
> Cited the paper in the revised manuscript.

- **New Results Asked:** InferSpec on non-mathematical reasoning domains
> New results are provided.

# Reviewer v1Ds:

The reviewer highlighted several strengths of our submission, noting that the high inference cost of LLM reasoning is an important problem, and that our motivation of using lightweight, model-internal signals is both practical and well-justified. The reviewer requested new results and clarifications as detailed below.

- **New Results Asked:** Performance of Self-Consistency Selector and LPBV as a verifier?
> New results and further clarifications are provided (Line-402-404 in Revised Manuscript).

- **New Results Asked:** Quantitative analysis demonstrating ensemble score correlates with ground-truth step-wise correctness
> Already included a quantitative step-wise analysis in the Appendix (Tables 6–9)

We are encouraged by the outcome of the rebuttal phase, and we have addressed all the concerns raised by the reviewers. The constructive nature of the reviews has helped us further strengthen our work, both in terms of technical rigor and presentation. We believe the revised manuscript reflects these improvements and makes a strong contribution to the field.

Thank you,

The Authors

---

### Note · Authors · 2026-01-26

I have read and agree with the venue's withdrawal policy on behalf of myself and my co-authors.

---

### Meta-Review · Area_Chair_SAT4 · 2026-01-10

**Summary:**

This paper proposes InferSpec, a verifier-guided speculative decoding framework for multi-step reasoning that replaces external reward models with model-internal verification signals: an attention-rollout–based grounding score (ABGV) and a log-probability–based confidence score (LPBV), combined with self-consistency selection to adaptively allocate compute between a draft and a target model. The submission’s strongest aspect is a practical motivation and a coherent pipeline that aims to improve reasoning reliability while retaining inference efficiency.

That said, I recommend rejection because the current evidence does not yet justify the paper’s broader claims about grounded step verification and generality. The core premise is that ABGV and LPBV are meaningful proxies for step-wise correctness, but the validation remains indirect: the rebuttal points to appendix analyses and qualitative contrasts with PRM behavior, and adds an ablation (SC+LPBV vs InferSpec) suggesting ABGV contributes, yet it still falls short of establishing robust proxy validity (e.g., calibrated quantitative correlation and failure-mode characterization across settings). Empirically, the scope remains narrow: results are dominated by math/formal reasoning benchmarks, with only a limited additional non-math evaluation (NL2SQL) introduced in rebuttal, leaving it unclear whether the attention-grounding heuristic transfers reliably to domains with different attention and confidence dynamics. Finally, the efficiency narrative is somewhat confounded by the multiple interacting components (multi-sample self-consistency, thresholding/ensemble weighting, and occasional multi-sampling with the target), making it hard to attribute gains cleanly to the proposed verifier and to assess how sensitive results are to tuning and overhead in realistic deployments. Overall, the work is promising and may become strong with more foundational validation and broader evaluation, but it is not yet at the level of rigor expected for acceptance.

**Reviewer Concerns:**

The rebuttal partially addressed reviewer requests for missing baselines and clarifications. In particular, it adds a target-only majority voting baseline that shows InferSpec can match or slightly exceed target-only majority accuracy while using fewer target calls (helpful for positioning), and it clarifies the memory overhead claim by stating they store attention from only the last layers and sparsify attention. It also adds an ablation comparing SC+LPBV to the full method, which strengthens the case that ABGV adds value beyond confidence-only verification.

However, several concerns remain outstanding. The most important is the foundational validation of ABGV/LPBV as correctness proxies: the response relies primarily on appendix tables and illustrative cases rather than a direct, robust quantitative validation that would convince a skeptical reader the verifier generalizes and is not a domain-specific heuristic. The generality concern is only weakly addressed: adding NL2SQL is a step in the right direction, but the evidence still does not support broad claims across reasoning domains (e.g., code, multi-hop QA, more open-ended tasks) or across different prompting regimes. There is also residual uncertainty about fairness of runtime comparisons and sensitivity to configuration, since the method’s performance depends on multiple sampling/selection and thresholding choices whose robustness and overhead are not fully stress-tested.

**Reviewer Scores:**

Reviewer adcc (rated 4, marginal below): potentially to increase modestly after seeing the added target-only majority baseline and clarifications on attention storage/sparsification and the rationale for multi-sampling with the target. I would expect a move to around 5–6 (weak accept / borderline) if they prioritize empirical improvements and the strengthened baseline story.

Reviewer GsNE (rated 4, marginal below): may increase slightly due to the added non-math result (NL2SQL) and citation/discussion of SpecReason, but their main concern about limited domain generalizability is only partially resolved. I would expect 5 (borderline) rather than a clear accept.

Reviewer v1Ds (rated 4, marginal below): may shift minimally upward because the rebuttal points to step-wise analyses in the appendix and adds the SC+LPBV ablation, directly addressing their “missing critical ablation” concern. Still, their central critique about lack of direct proxy validation and unclear utility beyond narrow domains is not fully settled, so I would expect them to remain at 4.

---

### Decision · Program_Chairs · 2026-01-26

Reject